# Development and Validation of Two Diagnostic Real-Time PCR (TaqMan) Assays for the Detection of *Bordetella avium* from Clinical Samples and Comparison to the Currently Available Real-Time TaqMan PCR Assay

**DOI:** 10.3390/microorganisms9112232

**Published:** 2021-10-27

**Authors:** Amro Hashish, Avanti Sinha, Amr Mekky, Yuko Sato, Nubia R. Macedo, Mohamed El-Gazzar

**Affiliations:** 1Department of Veterinary Diagnostic and Production Animal Medicine, College of Veterinary Medicine, Iowa State University, Ames, IA 50011, USA; hashish@iastate.edu (A.H.); asinhac@gmail.com (A.S.); ysato@iastate.edu (Y.S.); nubia@iastate.edu (N.R.M.); 2National Laboratory for Veterinary Quality Control on Poultry Production, Animal Health Research Institute, Agriculture Research Center, Giza 12618, Egypt; amrmekky@ahri.gov.eg

**Keywords:** *Bordetella avium* (BA), bordetellosis, TaqMan real-time PCR (qPCR), bacterial detection, clinical samples, analytical validation

## Abstract

*Bordetella avium* (BA) is one of many pathogens that cause respiratory diseases in turkeys. However, other bacterial species can easily overgrow it during isolation attempts. This makes confirming the diagnosis of BA as the causative agent of turkey coryza more difficult. Currently, there are two PCR assays for the molecular detection of BA. One is conventional gel-based PCR and the other is TaqMan real-time PCR (qPCR) assay. However, multiple pitfalls were detected in both assays regarding their specificity, sensitivity, and efficiency, which limits their utility as diagnostic tools. In this study, we developed and validated two TaqMan qPCR assays and compared their performance to the currently available TaqMan qPCR. The two assays were able to correctly identify all BA isolates and showed negative results against a wide range of different microorganisms. The two assays were found to have high efficiency with a detection limit of approximately 1 × 10^3^ plasmid DNA Copies/mL with high repeatability and reproducibility. In comparison to the currently available TaqMan qPCR assay, the newly developed assays showed significantly higher PCR efficiencies due to superior primers and probes design. The new assays can serve as a reliable tool for the sensitive, specific, and efficient diagnosis of BA.

## 1. Introduction

The genus *Bordetella* is comprised of fifteen different species [1,2]. *B. bronchiseptica, B. holmesii, B. parapertussis,* and *B. pertussis* are adapted to mammalian hosts and are phylogenetically closely related, while the more distantly related *B. avium* and *B. hinzii* are more associated with avian species [3].

*Bordetella avium* (BA) is a Gram-negative, non-fermentative, motile, aerobic bacilli [4] causing bordetellosis, also known as Turkey Coryza, in domesticated turkeys [5] and is an opportunistic pathogen in chickens [6]. Although mortality associated with uncomplicated bordetellosis in turkey is low, morbidity often approaches 100%, and infected turkeys are particularly susceptible to secondary bacterial infection [5]. In 2019, BA was ranked to be seventh in the list of the top health issues in the American turkey industry, consistently fluctuating between #5 and #8 for the past several years [7]. BA can also infect a variety of wild birds [8,9] and is associated with the Lockjaw Syndrome in *Psittacine* birds [10]. Moreover, a study by Harrington et al., in 2009 [11] provided some evidence of human respiratory infection with BA. *B. hinzii*, the closest genetic relative to BA, was referred to as *B. avium*-like before 1995 [12]. *B. hinzii* is frequently detected in the respiratory tracts of poultry as a commensal organism. Register and Kunkel [13] reported that some strains of *B. hinzii* have the potential to cause respiratory disease in turkeys.

Bacterial culture is the gold standard for laboratory diagnosis of bacterial diseases [14]. However, other bacterial species usually overgrow more fastidious microorganisms like BA. Due to limitations of direct culture and biochemical identification, detection of bacterial etiologies is often performed by Polymerase Chain Reaction (PCR), which is faster and more sensitive [15,16,17,18,19]. Currently, there are two published PCR assays developed for the diagnosis of BA [9,18,19]. The first assay is a conventional gel-based PCR (cPCR), initially developed by Savelkoul et al. [19], and then further optimized by Register and Yersin [18]. The limit of detection of this assay was stated to be between 15 pg and 20 pg—corresponding to 3750 to 5000 genome copies, which makes the assay useful only to confirm suspected bacterial colonies but not recommended for the detection of BA directly from clinical samples [18]. Regarding the specificity, the assay’s primers showed cross-reactivity with one *Staphylococcus hyicus* isolate, with a weak but reproducible non-specific band, revealing the specificity of the assay equal to 98.8%.

The other PCR for BA is a TaqMan-based real-time PCR (qPCR) that was designed to target a more species-specific gene (*rec*A) [9]. However, basic in-silico analysis of this assay showed a significant deviation from the general concepts for PCR primer and probe design [20]. The assay’s Efficiency (E) was stated to be as low as 70.1%, with a coefficient of determination (R^2^) equalling 0.953. Given all the mentioned limitations for both available assays, poultry diagnosticians are left with no reliable qPCR assays to confirm the diagnosis of BA in cases of respiratory disease in turkeys.

In this manuscript, we describe the development and validation of two TaqMan qPCR assays and compare the two assays’ performance to the currently available TaqMan qPCR assay. The newly developed assays would serve as a much-needed and reliable diagnostic tool that is sensitive, specific, and efficient in confirming the diagnosis of BA.

## 2. Materials and Methods

### 2.1. Target Genes Selection

To determine a specific target region for our primers and probes within the BA genome, we reviewed the full genome sequence analysis of BA (strain 197 N) published by Sebaihia et al. [2]. This genome comprises 3,732,255 base pairs and has limited synteny with other *Bordetella* genomes. The number of total genes within the BA genome (strain 197 N) is 3431 total genes. After reviewing the full genome of BA, we selected four unique genes (BAV1945, *fha*C, *hagA*1, and *hagB*2) as potential targets to develop our TaqMan qPCR assays (Table 1). Using Basic Local Alignment Search Tool (BLAST) [21] of these selected genes against NCBI BLAST nt database [21,22] indicated these genes sequences are specific only to BA. One more BLASTing of these genes was done against all available genome sequences of the genus *Bordetella* at the GenBank to exclude any high similarity of these genes with other species within the genus *Bordetella*.

All available sequences (*n* = 24) of these four genes were downloaded from BA genomes at GenBank, then all available sequences from each gene were aligned using CLUSTAL W [23] within MEGA X software [24]. A 900 bp conserved segment among all BA genomes was chosen within each of the four genes. This 900 bp segment for each gene was then BLASTed again against NCBI BLAST nt database [21,22] to confirm its specificity only to BA. Subsequently, a pair of primers were designed for each selected segment using Primer-BLAST [25] to amplify each of these four 900 bp targets from five selected clinical BA isolates included in this study.

PCR amplification of these four targets resulted in successful and consistent amplification of only two targets. The 900 bp segments within BAV1945 and *fha*C genes were selected as potential targets to design the primers and probes for two TaqMan qPCR assays.

### 2.2. Primers and Probes Design and Reaction Conditions

Primers and TaqMan^®^ probes were designed within the two selected targets (900 bp) of BAV1945 and *fha*C using Primer3Plus web interface [26] according to the general concepts for PCR primers and probe design [27]. The designed primers and probes were tested for their specificity through in-silico analysis using the BLAST search tool [21]. They were then analyzed using the online IDT oligo Analyzer 3.1 tool (https://www.idtdna.com/calc/analyzer (accessed on 16 January 2019)) to test for the potential formation of secondary structure and primer dimers. All oligonucleotides (primers and probes) were synthesized by IDT (Integrated DNA Technologies, Coralville, IA, USA). The primers and probes’ sequences for each of the two assays and their parameters are displayed in Table 2.

### 2.3. Real-Time PCR Conditions

The two assays were conducted under similar reaction conditions. Primers and probe were utilized in a 20 µL reaction containing 5 µL of TaqMan Fast Virus 1-step Master Mix (Applied Biosystems, Carlsbad, CA, USA), primers to a final concentration of 0.4 µmol, probe to a final concentration of 0.2 µmol, 8.135 µL of water, and 5 µL of DNA template.

Each reaction was conducted in Real-Time PCR System 7500 (Applied Biosystems, Carlsbad, CA, USA). Based on the calculated T_m_ of the primers and probes, the following amplification conditions were adopted: 50 °C for 5 min; 95 °C for 20 s with optics off; 40 cycles of 95 °C for 15 s followed by 60 °C for 60 s with optics on.

A non-template control (PCR-grade H_2_O) and positive control (isolated DNA of BA isolate confirmed by matrix-assisted laser desorption ionization-time of flight mass spectrometry (MALDI-TOF) were included in each run. All results were analyzed using SDS 1.5.1 software (Applied Biosystems, Carlsbad, CA, USA).

The primers and probe of the currently available TaqMan qPCR assay (*rec*A assay) [9] were ordered from IDT (Integrated DNA Technologies, Coralville, IA, USA) (Table 2). Based on the calculated T_m_ of the primers and the probe, the assay was conducted under the same reaction conditions of the two developed assays using the same Master Mix kit, the same primers, and probe concentrations, and the same thermal cycling conditions mentioned above.

### 2.4. Bordetella Avium Isolates and Clinical Samples

Twelve BA isolates were obtained from the Bacteriology section at the Iowa State University, Veterinary Diagnostic Laboratory (ISU-VDL), (Ames, Iowa) (Table 3). These isolates were cultured directly on blood agar and incubated for 24–48 h under aerobic conditions at 37 °C. Small, translucent colonies on blood agar characteristic of BA were confirmed by MALDI-TOF [18]. At the same time, three known BA-positive clinical samples (Table 3) were obtained from ISU-VDL. Subsequent confirmation of these three clinical samples was done through bacteriological isolation and confirmation by MALDI-TOF. Seventeen samples from BA negative flocks were also included, using homogenate of tracheas and lungs from apparently normal chickens and turkeys.

### 2.5. Other Bacteria and Viruses

Thirty-eight microorganisms (thirty-two bacterial pathogens and six viruses) that are likely to be found in samples submitted for BA diagnosis were also examined as controls to analyze the qPCR assays’ exclusivity. All microorganisms included in this study and their growth conditions are listed in Table 3.

### 2.6. Nucleic Acid Extraction

Nucleic acids were extracted using 100 µL from all bacterial or viral isolates’ suspension and 100 µL from swab suspension or tissue homogenate originated from BA clinical samples.

Nucleic acid extraction was conducted using a MagMAX™ Pathogen RNA/DNA Kit (Thermo Fisher Scientific, Waltham, MA, USA) on a Kingfisher-Flex instrument (Thermo Fisher Scientific) following the instructions of the manufacturer. Nucleic acids were eluted into 90 μL of elution buffer.

### 2.7. Evaluation of qPCR Assays’ Performance

#### 2.7.1. In-Silico Validation and Evaluation of the Primers and Probes

All primers and probes included in this study, including previously published assay, were in-silico tested for specificity using the BLAST search tool [21]. They were also analyzed using the online IDT oligo Analyzer 3.1 tool (https://www.idtdna.com/calc/analyzer (accessed on 16 January 2019)) to analyze the presence of any secondary structure and primer dimers.

#### 2.7.2. Analytical Validation and Evaluation of the qPCR Assays

Analytical validation and evaluation of the three assays were performed according to guidelines and previously published work [18,41,42,43,44,45,46,47] in order to confirm that the qPCR assays will perform reliably and consistently when implemented in a diagnostic setting. The following parameters were included:

A. Analytical Specificity: to measure the reactivity of the assays against our target (BA) and non-target microorganisms. Analytical Specificity comprises inclusivity and exclusivity: Inclusivity is the ability of each qPCR assay to detect different isolates of BA. This was evaluated by testing the assays against twelve BA bacterial isolates. Exclusivity is the lack of positive results from non-target pathogens. This was evaluated by running the assays against a panel of RNA or DNA extracts from thirty-eight different isolates (thirty-two bacterial and six viral) known to normally inhabit or infect the avian respiratory tract (Table 3). This list of microorganisms included a large number of *B. hinzii* isolates (*n* = 20) due to their close relationship to BA. Additionally, clinical respiratory samples from apparently normal birds are used to test for possible cross-reactivity of the developed assay with any normal flora in the respiratory system (Table 3).

B. Diagnostic specificity against clinical samples: through testing of the three assays against a number of known positive and known negative clinical samples submitted for BA diagnosis. Diagnostic specificity = True negatives/(true negatives + false positives) × 100.

C. Limit of detection (Analytical sensitivity): is the lowest copy number that each assay can reliably detect to determine the presence or absence of BA in a sample. This was performed through:

C.1 Construction of *Bordetella avium* BAV1945, *fhaC* and *rec*A positive control DNA: GBlock, which is a double-stranded synthetic DNA fragment, for the BAV1945 sequence of 323 bp, *fhaC* sequence of 306 bp and *rec*A sequence of 250 bp length, containing the forward, reverse primers and probe sequences, were ordered from IDT (Integrated DNA Technologies, Coralville, IA, USA). The inserts were cloned into pCR^®^ -Blunt II TOPO^®^ (Invitrogen™) using the manufacturer’s recommendations. Briefly, GBlock inserts were rehydrated in Tris-EDTA buffer (Invitrogen™) to make a concentration of 25 ng/µL. Four µLs of each GBlock suspension was mixed separately with 1 µL pCR^®^ -Blunt II TOPO^®^ vector and 1 µL salt solution and left at room temperature for 5 min for ligation. A three µL ligation mix from each reaction was then used to transform one-shot TOP10 chemically competent cells (Invitrogen™) and grown overnight in ampicillin agar plates separately at 37 °C. A white colony was picked from each plate the next day and grown overnight in ampicillin broth, after which the plasmids were extracted using QIAprep Spin Miniprep Kit (QIAprep^®^). All of the three plasmids were sequenced to confirm the presence of the inserts and subsequently converted to copy numbers after being quantified with the Nanodrop^TM^ spectrophotometer (ND1000 Thermo Scientific) using the following equation:$$\mathrm{Number}\;\mathrm{of}\;\mathrm{copies}\;=\frac{X\;\mathrm{ng}\;*\;6.0221\times 10^{23}\;\mathrm{molecules}/\mathrm{mole}}{\left(N\;*\;\frac{660\;g}{\mathrm{mole}}\right)\;*\;\;1\;*10^{9}\;\;\mathrm{ng}/g}$$
where: X = Nanodrop read (ng). N = length of the insert. 660 g/mole = average mass of one bp dsDNA, then stored at −80 °C until further use.

C.2. Generation of standard curves and estimation of the limit of detection for the three assays: Ten-fold serial dilutions of the constructed positive control DNA for the three assays containing (1 × 10^11^–1 × 10^1^) copies/mL were made to generate the standard curve, and 5µL was used as a template in each reaction.

The limit of detection was estimated using average Threshold Cycle (C_T_) values obtained from three independent qPCR runs, with each run containing four replicates. The average C_T_ values were plotted against log_10_ of ten-fold serial dilutions of plasmid DNA (copy number/mL), and linear equations were generated with R^2^ values for the three targets.

D. Efficiency (E): This is expressed in percentage and indicates the fraction of target molecules copied in one PCR cycle. The assays’ overall efficiency was estimated using the standard curve slope, as shown in the following equation: Efficiency = {10(−1/slope) − 1} × 100

E. Coefficient of determination (R^2^): R^2^ is the square of the Pearson’s correlation coefficient (r). It refers to how well the defined C_T_ values correlate with the dilution series. R^2^ gives an indication of the consistency of serial dilutions and pipetting errors.

F. Dynamic range: The range between the highest and lowest detectable copy number within the standard curve for which acceptable linearity (R^2^ ≥ 0.98) and efficiency (between 90–110%) are observed.

G. Repeatability (Intra-assay variation): To evaluate the repeatability, every single qPCR run for each assay contained four replicates of each ten-fold serial dilution. The repeatability was then analyzed based on the standard deviation (SD) and the coefficient of variability (CV) of the C_T_ average.

H. Reproducibility (Inter-assay variation): To evaluate the reproducibility, each dilution of the standard curve for the three assays was tested in four independent qPCR runs. All validation runs were performed on different days. The reproducibility was then analyzed based on the standard deviation (SD) and the coefficient of variability (CV) of the average C_T._

## 3. Results

Two new TaqMan probe-based qPCR assays for the detection of BA were developed and validated. Thereafter, these two assays were compared to the currently available TaqMan qPCR assay (*rec*A assay) [9].

### 3.1. Primers and Probe Design and Reaction Conditions

Two genes (BAV1945 and *fha*C) were selected to act as a target for designing the primers and probes for the two qPCR assays. For the BAV1945 gene, forward and reverse primers were designed to amplify 80 bp segment from nt number 2,072,515 to 2,072,594 (numbering according to accession number AM167904). The probe for this assay was designed to anneal 14 nucleotides away from the 3′ end of the forward primer. On the other hand, primers for *fha*C gene were designed to amplify a segment of 114 bp from nt number 2,095,559 to 2,095,672 (numbering according to accession number AM167904). The probe was designed to anneal nine nucleotides away from the reverse primer, as shown in Table 2. The difference in melting temperature (T_m_) of the forward and reverse primers within each assay was less than 1 °C. Simultaneously, the probes showed T_m_ (6–8 °C) higher than the primers.

### 3.2. In-Silico Validation and Evaluation of the Primers and Probes

The in-silico analysis through BLAST of the primers and probes of the two assays developed in this study (BAV1945 and *fhaC* assays) did not display high similarity to any of the GenBank sequences other than BA sequences, revealing high specificity. This is confirmed by one hundred percent of identity and query cover for our primers and probes with BA sequences within the Genbank. Additionally, using the online IDT oligo Analyzer tool revealed no significant primer dimer formations (self or hetero-dimer) or primer/probe dimer formation. On the other hand, in-silico analysis of the primers and probe of the currently available assay (*rec*A) showed that the forward and reverse primers are only 17 and 15 nts in length, leading to the low specificity during the BLASTing analysis. The GC content of forward and reverse primers of this assay showed higher than standard content for both primers (70.6% GC and 86.7%, respectively). By checking primer dimer formation, several significant secondary structures were revealed. For example, extensible reverse primer self-dimer and extensible reverse primer and probe hetero-dimer. In addition, there was a strong non-extensible hetero-dimer between forward and reverse primer and a strong non-extensible probe self-dimer. Additionally, the forward and reverse primers showed more than 4 °C differences in T_m_. Moreover, the probe showed a lower T_m_ (5–10 °C) than the primers.

### 3.3. Analytical Validation and Evaluation of the qPCR Assays

Analytical specificity (Inclusivity and Exclusivity): All three assays, including the currently available *rec*A assay, were able to correctly identify the tested twelve strains of BA. Additionally, they all showed 100% specificity against the panel of microorganisms, including the closely-related *B. hinzii*, likely to be found in samples submitted for BA diagnosis (Table 3).Evaluation of the assays’ diagnostic specificity against clinical samples: All three assays showed 100% diagnostic specificity. All assays were able to detect only the *Bordetella avium* positive known clinical samples with no cross-reactivity against clinical samples from apparently normal birds.Limit of detection: The limit of detection, the lowest concentration of analyte at which 95% of samples for that concentration are classified as positive, was calculated for the three assays by plotting average C_T_ values from three independent runs against log_10_ of 10-fold serial dilutions (10^11^–10^3^) of plasmid DNA (copy number/mL). Despite showing different amplification efficiencies, the limit of detection for the three assays was the same (approximately 1 × 10^3^ plasmid DNA Copies/mL) as shown in Table 4.Efficiency: Using the slope from the linear equation which was generated from the standard curve, the overall efficiency was estimated to be 101.32% (for the BAV1945 assay) and 105.89% (for the *fha*C assay), which is within the acceptable range (90–110%). On the other hand, the amplification efficiency for the *rec*A assay (122.16%) exceeded that acceptable range. (Table 4 and Figure 1 and Figure 2).Coefficient of determination (R^2^): Plotting average C_T_ values from three independent runs against log10 of 10 fold serial dilutions (from 10^11^–10^3^) of plasmid DNA (copy number/mL) of the three assays generated a linear equation with R^2^ equals (0.999) and (0.998) for the newly designed assays. At the same time, the *rec*A assay has an R^2^ equals (0.995). R^2^ > 0.98 is acceptable for well-designed qPCR assays which indicates the consistency of serial dilutions.Dynamic range: The newly developed assays showed a wide dynamic range (from C_T_ 7.81 to C_T_ 34.15) while maintaining amplification linearity of at least nine orders of magnitudes (Table 5 and Figure 1). On the other hand, the dynamic range of the *rec*A assay could not be determined due to the efficiency of the assay having a value (E = 122.16%) over the acceptable limit (E > 110%).Repeatability: The intra-assay coefficient of variability (%CV) for the C_T_-values determined for the BAV1945 assay ranged from (0.01–1.32%) with an average = 0.43%, while the %CV for the *fha*C assay ranged from (0.06–1.2%) with an average = 0.61%. On the other hand, the %CV for the *rec*A assay ranged from (0.21–1.53%) with an average = 0.89% (Table 5). These values demonstrate the good repeatability of all three assays (%CV less than 10% is acceptable for intra-assay variability [48]).Reproducibility: The inter-assay %CV for the C_T_-values determined for the BAV1945 assay ranged from (0.19–1.39%) with an average = 0.53%, while the %CV for the *fha*C assay ranged from (0.1–1.47%) with an average = 0.55%. On the other hand, the inter-assay %CV for the *rec*A qPCR ranged from (0.61–4.43%) with an average = 2% (Table 5). These values reveal the acceptable reproducibility of all three assays (%CV less than 15% is acceptable for inter-assay variability [48]).

## 4. Discussion

The use of qPCR as a diagnostic assay has become widely popular due to its speed, better sensitivity, specificity, and reduced risk of carryover contamination compared to conventional diagnostic methods [49,50]. Numerous qPCR assays have been developed for the diagnosis of poultry pathogens [51,52,53,54,55], including one assay for the diagnosis of BA using the *rec*A gene as a conserved and specific target [9,18,19].

While validating this assay for its use in ISU-VDL, we discovered potential issues with its diagnostic accuracy. The in-silico analysis of the primers and probe revealed a short length of both primers (15 and 17 nts). PCR primers are the main determinants of PCR specificity. Shorter primers (<17 nts) may decrease the specificity of the reaction [56]. To ensure the primers’ specificity and prevent annealing to any non-specific targets, primers are usually designed to be in the order of 18–24 nucleotides in length [42,57]. These short-length primers affected the in-silico specificity analysis. During the in-silico analysis, in addition to BA, various microorganisms presented identity and query coverage equal to 100%. In addition to the primers, the probe was also very short (14 nts in length). However, a “No significant similarity found” message was shown during BLAST in-silico specificity analysis with all sequences in GenBank.

Additionally, there was a high GC content in the forward (70.6%) and in the reverse (86.7%) primers. The recommended GC content in primers is to be between 40–60% of a primer sequence [58] to ensure stable but specific binding of primer and template and allow efficient annealing [20] without the promotion of non-specific annealing or secondary structure formation. Additionally, the reverse primers showed high GC content at the 3′ end (Table 2). A high GC content at the 3′ end of a primer may reduce the reaction’s specificity by allowing extension after non-specific annealing of the 3′ end, even if there is no complete annealing of the remainder of the primer sequence [59]. Another limitation of this probe design is the high number of G nt (6/14) within the sequence. It is preferred that the qPCR probe contain more C than G because such probes produce a greater normalized change in fluorescence (∆ Rn). Easier interpretation of the results is obtained with larger ∆ Rn, as low positive signals can be more easily differentiated from background signals [42,60].

For all of the above-mentioned reasons, and additional reasons mentioned below, it was decided that implementing this *rec*A qPCR assay will not yield reliable diagnostic results and it is necessary to develop a new qPCR assay for the diagnosis of BA.

In this manuscript, we described the development and validation of two new TaqMan qPCR assays for the efficient, sensitive, and specific detection of BA. Initially, the intent was to select one of the two targets for the PCR assay. However, the analytical and clinical validation showed that the two targets are equally specific and sensitive to *Bordetella avium*. Then, we compared the performance of the two newly developed assays to the currently available TaqMan qPCR assay.

The primer and TaqMan probe sets designed in our assays target two genes (BAV1945 and *fhaC*), identified as unique BA genes in the complete genome analysis of BA (strain 197 N) [2]. The selected targets, as well as the designed primers and probes sequences, were analyzed for their specificity by in-silico analysis using the BLAST tool, which revealed no similarity to any of the GenBank sequences other than *Bordetella avium* sequences, including *B. hinzii*, the closest species to BA [12].

Oligo-dimers are off-target amplification artifacts formed by primer-primer or primer-probe binding. These interactions can competitively reduce binding to target DNA and exhaust deoxynucleotides—all of which result in reduced amplification efficiency and suboptimal product yields [46]. The in-silico analysis of the selected primers and probes sequences of the newly developed assays revealed the absence of any significant primer-dimer formation or primer-probe heterodimer formation. The lack of dimer formation leads to better amplification efficiency.

The in-silico analysis of the currently available *rec*A assay revealed several significant secondary structures, either extensible, e.g., extensible reverse primer self-dimer, or non-extensible, e.g., strong non-extensible probe self-dimer. Extensible dimers are formed when the 3′ end of an oligo is perfectly complementary to another oligo. These extensible dimers serve as templates for the DNA polymerase and can lead to off-target amplification. On the other hand, the presence of non-extensible dimers may affect the efficiency of the assay by decreasing the number of primer or probe molecules available for binding to the target gene [18,61].

To avoid false hybridization of primers, the difference in T_m_ of forward and reverse primers preferred to be less than 1 °C [56]. However, the difference in T_m_ between the forward and reverse primer of the *rec*A assay was more than 4 °C. Additionally, the designed probe of the *rec*A qPCR assay had lower T_m_ than the primers. For standard qPCR assays, it is recommended that the T_m_ of the probe should be at least 5–10 °C degrees higher than the T_m_ of the primers [62]. Probe with T_m_ lower than those of the primers will anneal to the target after the primers; subsequently, the polymerase may begin amplification of the target that does not contain bound probe. As a result, synthesis of the DNA might begin without associated probe degradation, and thus, an increase in fluorescence will not be detected. Such a situation leads to the generation of inaccurate data. The difference in T_m_ between forward and reverse primers in the new assays was 1 °C, while the T_m_ of the probes for the newly developed assays was 6 °C higher than the T_m_ of the primers.

Establishing performance characteristics, such as analytical specificity, diagnostic specificity, the limit of detection, dynamic range, repeatability, and reproducibility, are fundamental for the analytical evaluation of any qPCR assay [41,42,45,63].

Both the newly designed assays were found to be highly specific; they were able to inclusively detect different BA isolates with the absence of any cross-reactivity to any other tested microorganism (either taxonomically related e.g., *B. hinzii* and *B. bronchoseptica* or pathogens that are likely to be found in samples submitted for BA diagnosis) (Table 3). Unexpectedly, high specificity was also shown for the *rec*A assay, despite the poor primers and probe design and very low specificity during the in-silico analysis. On the other hand, diagnostic specificity was 100% for the two newly developed assays using confirmed negative and positive BA clinical samples. This also was unexpectedly true for the *rec*A assay.

The detection limit for both of the newly designed assays (≈1000 copies/mL, which is ≈five copies per reaction) reveals high analytical sensitivity. The detection limit for the *rec*A assay was also ≈1000 copies/mL, indicating it is as sensitive as the two newly developed assays.

Describing key quality control parameters such as efficiency and the R^2^ is essential for correct interpretation of qPCR results as stated in the Minimum Information for Publication of Quantitative Real-time PCR Experiments (MIQE) guidelines [41]. Coefficient of determination (R^2^) is used to assess the linearity of the data [64] and to indicate the variability of the assay replicates. R^2^ should be close to 1 (higher than 0.98). This study’s two developed assays showed R^2^ ≥ 0.998, and the R^2^ for the *rec*A assay was ≥0.994. This indicates a good correlation between the amount of template and the C_T_ values [42]. R^2^ ≥ 0.994 for the three assays also eliminates the possibility of pipetting or dilution errors, which indicates the accuracy and reliability of the results.

The PCR efficiency is one of the most critical indicators of the performance of a qPCR assay [65]. Ideally, qPCR efficiency should equal 100%. This means that the PCR product will be doubled with each round of amplification [66]. Efficiency was calculated to be 101.32% and 105.89%% for the BAV1945 and *fha*C assays respectively, which is within the acceptable limit for an efficient qPCR assay [67].

Despite having a detection limit equal to 1000 copies/mL, the reliability of the *rec*A assay sensitivity could not be guaranteed. This is due to the assay’s unusually higher efficiency percentage (E = 122.16%). qPCR efficiency over the acceptable 110% limit can be potentially explained by the presence of some polymerase inhibitors e.g., carryover material in the sample or excessive amounts of DNA. Inhibition means that even when more template is added, the CT values are not shifted earlier cycles as expected. This will flatten the efficiency plot and lower the slope, so the efficiency goes high [65]. However, high reaction efficiency (>110%) is generally the result of primer–dimers or non-specific amplicons [68]. This might be explained by the poor design of the primers and probe of the assay along with the low specificity of the primers as evident during the in-silico analysis.

This study’s two developed assays showed a wide dynamic range of at least nine orders of magnitude with acceptable linearity (R^2^ ≥ 0.98) and high efficiency. On the other hand, the dynamic range could not be concluded from the *rec*A assay due to the reaction efficiency exceeding the acceptable limit.

To be implemented as a reliable diagnostic test, the qPCR assay should be repeatable and reproducible. The two developed qPCRs assays showed a good level of repeatability and reproducibility even at the highest dilutions. Intra-assay and inter-assay %CV of both assays were comparable to the *rec*A qPCR. The *rec*A assay showed acceptable sensitivity and specificity, but signs of primer–dimers or non-specific amplicons and abnormally high efficiency due to poor primer and probe design compromised the overall performance of the test.

## 5. Conclusions

In conclusion, qPCR design is usually relatively straightforward and it takes less time to design an efficient, sensitive, and specific assay than to troubleshoot a poorly designed one.

We have developed two efficient TaqMan-based qPCR assays for the detection of *Bordetella avium* from both bacterial isolates and clinical samples. Initially, the intent was to select one of the two targets for the PCR assay. However, the analytical and clinical validation showed that the two targets are equally specific and sensitive to *Bordetella avium*. Either assay can be used individually. These assays as diagnostic tools can improve the differential diagnosis of avian respiratory diseases, particularly in turkeys.

## Figures and Tables

**Figure 1 microorganisms-09-02232-f001:**
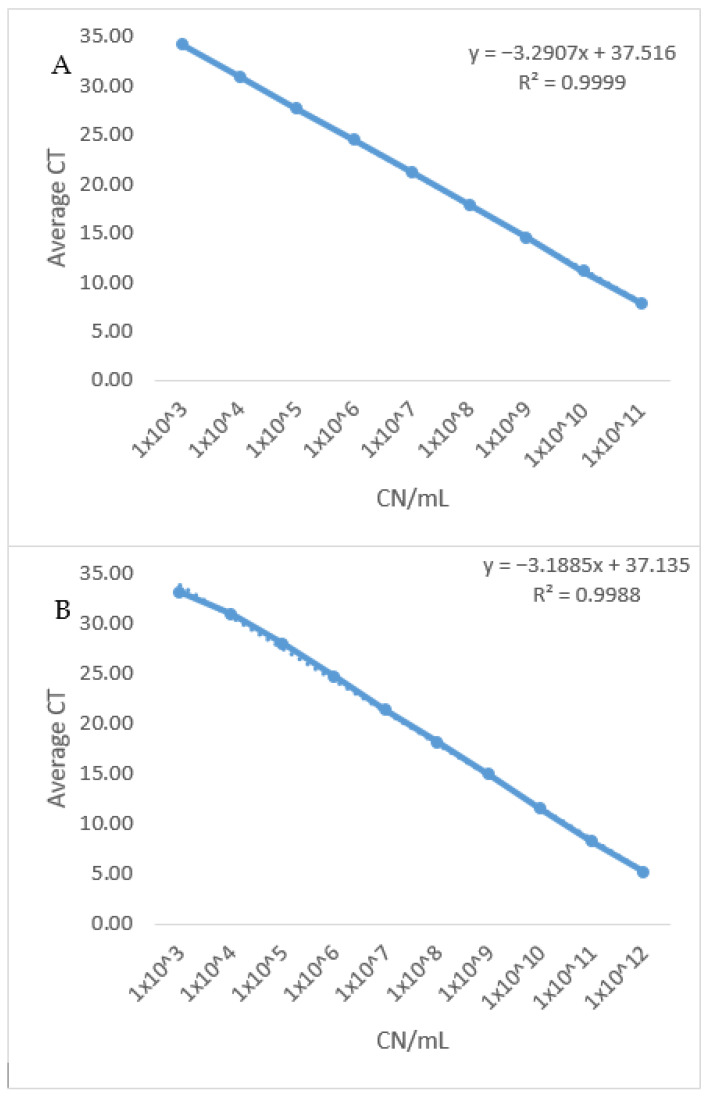
Standard curve of the two newly developed qPCR assays. (**A**) The standard curve of BA BAV1945 qPCR assay was generated by plotting average CT values from three independent runs against log10 of 10 fold serial dilutions (from 10^11^–10^3^) of plasmid DNA (copy number/mL). Reaction efficiency of 101.3% was estimated using the standard curve slope. (**B**): Standard curve of BA *fhaC* qPCR assay was generated by plotting average CT values from three independent runs against log10 of 10 fold serial dilutions (10^11^–10^3^) of plasmid DNA (copy number/mL). Reaction efficiency of 105.89% was estimated using the standard curve slope.

**Figure 2 microorganisms-09-02232-f002:**
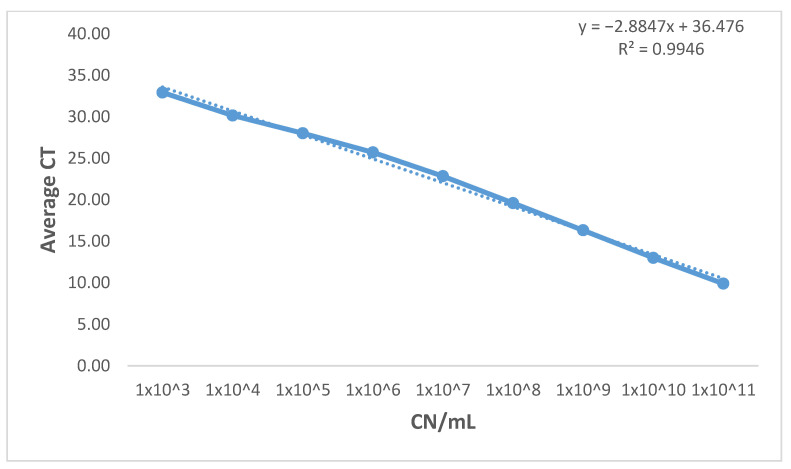
Standard curve of the currently existing (*recA*) qPCR. The standard curve of BA recA qPCR assay was generated by plotting average CT values from three independent runs against log10 of 10 fold serial dilutions (from 10^11^–10^3^) of plasmid DNA (copy number/mL). Reaction efficiency of 122.16% was estimated using the standard curve slope.

**Table 1 microorganisms-09-02232-t001:** Primary selected four *Bordetella avium* novel genes.

Gene Name	Locus Tag	Nucleotide Position *	Proposed Function
BAV1945	BAV1945	2,059,011–2,078,393	Putative Adhesion
*fha*C	BAV1961	2,094,955–2,096,733	Hemolysin Activator Protein
*hagA1*	BAV2824	3,062,685–3,064,298	Putative Hemolysin/hemagglutinin accessory protein
*hagB2*	BAV2819	3,055,039–3,059,094	Putative Hemolysin/hemagglutinin

* Nucleotide position according to accession number AM167904.1.

**Table 2 microorganisms-09-02232-t002:** Genomic targets and oligonucleotides characteristics of the three BA TaqMan qPCR assays included in this study.

Target Genomic Region	Oligo	Sequence (5′ to 3′)	No of (bp)	Nt Position *	Amplified Segment Length	Reference
BAV1945	F * Primer	GCC ACA ATC TCT TTA GCC TGA	21	2,072,515–2,072,535	80 bp	New (This study)
	R ** Primer	CTG GAA GAC AGC AAT AGC C	19	2,072,576–2,072,594		
	Probe	FAM CGT CAC GCA TCG TCT CGC CA BHQ	20	2,072,549–2,072,568		
*fha*C	F Primer	TT GCT ATT GAC CGC CAA CAG	20	2,095,559–2,095,578	114 bp	New (This study)
	R Primer	TTT GAC TCG AAC GCT CTA CC	20	2,095,653–2,095,672		
	Probe	FAM AC TTC CCA GTT CAG CGT GTA TGG TGT BHQ	26	2,095,619–2,095,644		
*rec*A	F Primer	CGGTTCGCTGGGCTTGG	17	2,491,504–2,491,520	50 bp	[9]
	R Primer	CACGCGGCAGCCCGC	15	2,491,471–2,491,485		
	Probe	FAM CATCGCGCTGGGTG BHQ	14	2,491,489–2,491,502		

Nucleotide position according to GenBank accession number (AM167904). * F = Forward Primer. ** R = Reverse Primer.

**Table 3 microorganisms-09-02232-t003:** *Bordetella avium* and other avian pathogens were used in this study.

Sample No.	Pathogen	Information (Age, spp. and Year)	Sample Type	BAV1945 qPCR Assay	*fhaC* qPCR Assay	*recA* qPCR Assay	Acc. to
1	*Bordetella avium*	53 Weeks-Chicken-2017	Bacterial Isolate ^a^	+	+	+	
2	*Bordetella avium*	7 weeks-Turkey-2017	Bacterial Isolate ^a^	+	+	+	
3	*Bordetella avium*	3 weeks-Turkey-2017	Bacterial Isolate ^a^	+	+	+	
4	*Bordetella avium*	6 weeks-Turkey-2018	Bacterial Isolate ^a^	+	+	+	
5	*Bordetella avium*	6 weeks-Turkey-2018	Bacterial Isolate ^a^	+	+	+	
6	*Bordetella avium*	6 weeks-Turkey-2018	Bacterial Isolate ^a^	+	+	+	
7	*Bordetella avium*	6 weeks-Turkey-2018	Bacterial Isolate ^a^	+	+	+	
8	*Bordetella avium*	6 weeks-Turkey-2018	Bacterial Isolate ^a^	+	+	+	
9	*Bordetella avium*	14 weeks-Turkey-2019	Bacterial Isolate ^a^	+	+	+	
10	*Bordetella avium*	20 weeks-Chicken-2020	Bacterial Isolate ^a^	+	+	+	
11	*Bordetella avium*	Unknown age-Chicken-2020	Bacterial Isolate ^a^	+	+	+	
12	*Bordetella avium*	6.5 weeks-Turkey-2020	Bacterial Isolate ^a^	+	+	+	
13	Known positive ^D^	6.5 weeks-Turkey-2020	Tracheal Swab ^b^	+	+	+	
14	Known positive ^D^	6 weeks-Turkey-2020	Tracheal Swab ^b^	+	+	+	
15	Known positive ^D^	6.5 weeks-Turkey-2020	Tracheal homogenate ^b^	+	+	+	
16	Known negative ^D^	4.5 years-Chicken-2019	Lung homogenate ^b^	-	-	-	
17	Known negative ^D^	4.5 years-Chicken-2019	Tracheal homogenate ^b^	-	-	-	
18	Known negative ^D^	1 week-Turkey-2019	Tracheal homogenate ^b^	-	-	-	
19	Known negative ^D^	1 week-Turkey-2019	Lung homogenate ^b^	-	-	-	
20	Known negative ^D^	Unknown age-Turkey-2019	Lung homogenate ^b^	-	-	-	
21	Known negative ^D^	Unknown age-Turkey-2019	Lung homogenate ^b^	-	-	-	
22	Known negative ^D^	10 days-Turkey-2019	Tracheal homogenate ^b^	-	-	-	
23	Known negative ^D^	10 days-Turkey-2019	Tracheal homogenate ^b^	-	-	-	
24	Known negative ^D^	36 weeks-Chicken-2019	Lung homogenate ^b^	-	-	-	
25	Known negative ^D^	38 weeks-Chicken-2019	Lung homogenate ^b^	-	-	-	
26	Known negative ^D^	38 weeks-Chicken-2019	Tracheal homogenate ^b^	-	-	-	
27	Known negative ^D^	85 weeks-Chicken-2019	Tracheal homogenate ^b^	-	-	-	
28	Known negative ^D^	85 weeks-Chicken-2019	Lung homogenate ^b^	-	-	-	
29	Known negative ^D^	2 days-Turkey-2019	Lung homogenate ^b^	-	-	-	
30	Known negative ^D^	2 days-Turkey-2019	Tracheal homogenate ^b^	-	-	-	
31	Known negative ^D^	3 weeks-Turkey-2019	Lung homogenate ^b^	-	-	-	
32	Known negative ^D^	3 weeks-Turkey-2019	Tracheal homogenate ^b^	-	-	-	
33	*Mycoplasma gallisepticum*		Bacterial Isolate ^a^	-	-	-	[28]
34	*Mycoplasma iowae*		Bacterial Isolate ^a^	-	-	-	[29]
35	*Mycoplasma synoviae*		Bacterial Isolate ^a^	-	-	-	[28]
36	*Ornithobacterium rhinotracheale*		Bacterial Isolate ^a^	-	-	-	[30]
37	*Bordetella hinzii*		Bacterial Isolate ^a^	-	-	-	[13]
38	*Bordetella hinzii*		Bacterial Isolate ^a^	-	-	-	[13]
39	*Bordetella hinzii*		Bacterial Isolate ^a^	-	-	-	[13]
40	*Bordetella hinzii*		Bacterial Isolate ^a^	-	-	-	[13]
41	*Bordetella hinzii*		Bacterial Isolate ^a^	-	-	-	[13]
42	*Bordetella hinzii*		Bacterial Isolate ^C^	-	-	-	[13]
43	*Bordetella hinzii*		Bacterial Isolate ^C^	-	-	-	[13]
44	*Bordetella hinzii*		Bacterial Isolate ^C^	-	-	-	[13]
45	*Bordetella hinzii*		Bacterial Isolate ^C^	-	-	-	[13]
46	*Bordetella hinzii*		Bacterial Isolate ^C^	-	-	-	[13]
47	*Bordetella hinzii*		Bacterial Isolate ^C^	-	-	-	[13]
48	*Bordetella hinzii*		Bacterial Isolate ^C^	-	-	-	[13]
49	*Bordetella hinzii*		Bacterial Isolate ^C^	-	-	-	[13]
50	*Bordetella hinzii*		Bacterial Isolate ^C^	-	-	-	[13]
51	*Bordetella hinzii*		Bacterial Isolate ^C^	-	-	-	[13]
52	*Bordetella hinzii*		Bacterial Isolate ^C^	-	-	-	[13]
53	*Bordetella hinzii*		Bacterial Isolate ^C^	-	-	-	[13]
54	*Bordetella hinzii*		Bacterial Isolate ^C^	-	-	-	[13]
55	*Bordetella hinzii*		Bacterial Isolate ^C^	-	-	-	[13]
56	*Bordetella hinzii*		Bacterial Isolate ^C^	-	-	-	[13]
57	*Bordetella bronchiseptica*		Bacterial Isolate ^a^	-	-	-	[31]
58	*Bordetella bronchiseptica*		Bacterial Isolate ^a^	-	-	-	[31]
59	*Paturella multocida*		Bacterial Isolate ^a^	-	-	-	[32]
60	*Paturella multocida*		Bacterial Isolate ^a^	-	-	-	[32]
61	*Escherichia coli*		Bacterial Isolate ^a^	-	-	-	[33]
62	*Gallibacterium anatis*		Bacterial Isolate ^a^	-	-	-	[34]
63	*Erysipelas rhusiopathiae*		Bacterial Isolate ^a^	-	-	-	[35]
64	*Staphylococcus aureus*		Bacterial Isolate ^a^	-	-	-	[36]
65	Avian Paramyxovirus-1 (Newcastle Disease)		Viral Isolate ^a^	-	-	-	[37]
66	Avian Reovirus		Viral Isolate ^a^	-	-	-	[38]
67	Infectious Bronchitis		Viral Isolate ^a^	-	-	-	[39]
68	Infectious Bronchitis		Viral Isolate ^a^	-	-	-	[39]
69	Infectious Bronchitis		Viral Isolate ^a^	-	-	-	[39]
70	Infectious Laryngotracheitis		Viral Isolate ^a^	-	-	-	[40]

^a^ Bacterial and viral isolates were provided by the Bacteriology and Virology section, College of Veterinary Medicine, ISU (Ames, IA). ^b^ All clinical samples were provided by Veterinary Diagnostic Laboratory–ISU. ^C^
*B. hinzii* isolates were obtained from the National Animal Disease Center. ^D^ Known positive and negative samples for *Bordetella avium*. The table shows the specificity panel used for the validation of the qPCR assays along with obtained results using the two assays. No cross-reactivity was obtained with any tested agents other than BA.

**Table 4 microorganisms-09-02232-t004:** Detection limits, standard curves linear equasions, R2 and Efficency and results of the three qPCRs.

Target Gene	Amplicon Size	Limit of Detection	Linear Equation	R^2^	Efficiency
BAV 1945	80 bp	1000 copy/mL	y = −3.2907x + 37.516	R^2^ = 0.999	E = 101.32%
*fha*C	115 bp	1000 copy/mL	y = −3.1885x + 37.135	R^2^ = 0.999	E = 105.89%
*rac*A	50 bp	1000 copy/mL	y = −2.8847x + 36.476	R^2^ = 0.995	E = 122.16%

**Table 5 microorganisms-09-02232-t005:** Intra- and inter-assay variations in different concentrations of positive control DNA of the three qPCRs.

Copy Number/mL	Repeatability			Reproducibility			Dynamic Range
	Mean C_T_	SD	CV%	Mean C_T_	SD	CV%	
BAV1945 TaqMan qPCR assay							
1.0 × 10^3^	33.58	0.44	1.32	34.15	0.47	1.39	Wide dynamic range while maintaining amplification linearity of at least nine magnitudes (from C_T_ 7.81 to C_T_ 34.15).
1.0 × 10^4^	30.83	0.12	0.40	30.89	0.06	0.19	
1.0 × 10^5^	27.62	0.10	0.35	27.60	0.08	0.27	
1.0 × 10^6^	24.32	0.00	0.02	24.42	0.08	0.32	
1.0 × 10^7^	21.04	0.00	0.01	21.19	0.16	0.77	
1.0 × 10^8^	17.81	0.05	0.30	17.88	0.08	0.44	
1.0 × 10^9^	14.49	0.02	0.13	14.52	0.05	0.32	
1.0 × 10^10^	11.12	0.03	0.28	11.10	0.03	0.23	
1.0 × 10^11^	7.73	0.08	1.05	7.81	0.06	0.80	
*fha*C TaqMan qPCR assay							
1.0 × 10^3^	33.59	0.40	1.20	33.17	0.29	0.87	Wide dynamic range while maintaining amplification linearity of at least nine magnitudes (from C_T_ 8.22 to C_T_ 33.17)
1.0 × 10^4^	30.78	0.35	1.15	30.93	0.19	0.61	
1.0 × 10^5^	27.86	0.03	0.10	27.95	0.07	0.24	
1.0 × 10^6^	24.66	0.08	0.32	24.71	0.07	0.26	
1.0 × 10^7^	21.35	0.19	0.90	21.35	0.02	0.11	
1.0 × 10^8^	18.10	0.04	0.23	18.12	0.02	0.10	
1.0 × 10^9^	14.92	0.12	0.80	14.89	0.05	0.34	
1.0 × 10^10^	11.55	0.01	0.06	11.49	0.17	1.47	
1.0 × 10^11^	8.20	0.06	0.70	8.22	0.08	0.95	
*rec*A TaqMan qPCR assay							
1.0 × 10^3^	33.03	0.48	1.46	32.93	1.46	4.43	Could not be determined due to the higher efficiency of the assay (E = 122.20%) over the acceptable range (90–110%).
1.0 × 10^4^	30.95	0.07	0.23	30.16	0.84	2.79	
1.0 × 10^5^	28.63	0.12	0.44	28.02	0.71	2.53	
1.0 × 10^6^	26.13	0.06	0.21	25.71	0.48	1.87	
1.0 × 10^7^	23.13	0.13	0.55	22.84	0.33	1.45	
1.0 × 10^8^	19.68	0.23	1.19	19.60	0.22	1.12	
1.0 × 10^9^	16.48	0.21	1.25	16.32	0.21	1.28	
1.0 × 10^10^	13.19	0.15	1.15	13.00	0.26	1.97	
1.0 × 10^11^	9.94	0.15	1.53	9.90	0.06	0.61	

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
