# Peer review of "Development and Validation of Two Diagnostic Real-Time PCR (TaqMan) Assays for the Detection of *Bordetella avium* from Clinical Samples and Comparison to the Currently Available Real-Time TaqMan PCR Assay"

_microorganisms, 2021, doi:10.3390/microorganisms9112232_

Round 1
Reviewer 1 Report
Development and Validation of Two Diagnostic Real-Time PCR (TaqMan) Assays for the Detection of Bordetella avium from Clinical Samples and Comparison to the Currently Available Real-Time TaqMan PCR Assay is an interesting and well written manuscript. The study is well planned and executed.
Some minor comments to address:
L168 OIE guidelines include selectivity, inclusivity and exclusivity as part of the analytic performance, not the latter two as part of the first. Please correct.
L240-246 Guidelines for author must cancelled before the manuscript is submitted.
Table 4: 50 bp for racA.
L418 false hybridization
Author Response
Dear Reviewer 1,
We highly appreciate your comments, corrections and suggestions. It surely made our manuscript better. Please find below our point by point response to your review
Author Response to Reviewer’s Comments (ID microorganisms-1413507)
Reviewer 1:
- 1. L168 OIE guidelines include selectivity, inclusivity and exclusivity as part of the analytical performance, not the latter two as part of the first. Please correct.
It is corrected throughout the entire manuscript “with track changes”.
- L240-246 Guidelines for author must cancelled before the manuscript is submitted.
The six lines were deleted. This was an overlook.
- Table 4: 50 bp for racA.
“bp” abbreviation was added to the table.
- L418 false hybridization
The “H” letter was changed from uppercase to a lowercase letter.
Sincerely,
Mohammed El-Gazzar

Reviewer 2 Report
This is a simple but informative study by Hashish et al where they have developed a couple of diagnostic Real-Time PCR assays for detecting the presence of pathogenic Bordetella avium in clinical samples. The authors have also compared its efficacy to the currently available Real-Time TaqMan PCR Assay. Their tests were correctly able to detect B. avium isolates and was able to distinguish against other microbes. Their results suggest high efficacy, reproducibility and repeatability of their Real-Time assays compared to the currently available TaqMan PCR assays primarily due to their superior primers and probes design. This is a valuable addition to the existing literature and would help veterinary pathologists to diagnose B. avium in avian species. The background and methods are well-described, and the experimental results are analyzed clearly. I would advise the authors to revise their manuscript for misspelt words and phrasing of sentences.
Author Response
Dear Reviewer 2,
We highly appreciate your comments, corrections and suggestions. It surely made our manuscript better. Please find below our point by point response to your review
Author Response to Reviewer’s Comments (ID microorganisms-1413507)
Reviewer 2: I would advise the authors to revise their manuscript for misspelt words and phrasing of sentences.
A check of the whole manuscript was done to correct any misspelled words. At the same time, paraphrasing of some sentences was made to make it more clear. These changes were “Track changed” in the re-submitted version of the manuscript.
Sincerely,
Mohammed El-Gazzar
